# The Interplay of Endothelial P2Y Receptors in Cardiovascular Health: From Vascular Physiology to Pathology

**DOI:** 10.3390/ijms23115883

**Published:** 2022-05-24

**Authors:** Cendrine Cabou, Laurent O. Martinez

**Affiliations:** 1I2MC, University of Toulouse, INSERM, UPS, 31000 Toulouse, France; 2Department of Human Physiology and Physiopathology, Faculty of Pharmacy, UPS, 31000 Toulouse, France

**Keywords:** purinergic signaling, endothelial cells, shear stress, vasodilatation, transcytosis, intimal hyperplasia, atherosclerosis, high-density lipoprotein

## Abstract

The endothelium plays a key role in blood vessel health. At the interface of the blood, it releases several mediators that regulate local processes that protect against the development of cardiovascular disease. In this interplay, there is increasing evidence for a role of extracellular nucleotides and endothelial purinergic P2Y receptors (P2Y-R) in vascular protection. Recent advances have revealed that endothelial P2Y_1_-R and P2Y_2_-R mediate nitric oxide-dependent vasorelaxation as well as endothelial cell proliferation and migration, which are processes involved in the regeneration of damaged endothelium. However, endothelial P2Y_2_-R, and possibly P2Y_1_-R, have also been reported to promote vascular inflammation and atheroma development in mouse models, with endothelial P2Y_2_-R also being described as promoting vascular remodeling and neointimal hyperplasia. Interestingly, at the interface with lipid metabolism, P2Y_12_-R has been found to trigger HDL transcytosis through endothelial cells, a process known to be protective against lipid deposition in the vascular wall. Better characterization of the role of purinergic P2Y-R and downstream signaling pathways in determination of the endothelial cell phenotype in healthy and pathological environments has clinical potential for the prevention and treatment of cardiovascular diseases.

## 1. Introduction: An Overview of P2Y Receptors in Endothelium

The vascular system supplies nutrients, hormones, and oxygen to organs. Its homeostasis depends on coordinated action of the blood flow [1,2] and its components [3,4,5,6,7,8,9,10] on the vascular wall. Endothelial cells form the first line of cellular contact with blood, and they sense mechanical, metabolic, and immunological stimuli in their microenvironment through specific receptors. Their interactions with the blood and cellular components elicit biochemical responses in the endothelium that are critical for the control of key functions such as vascular tone and blood pressure, leucocyte recruitment during inflammation or immunosurveillance, and vascular permeability. These interactions can lead to protection or alteration of the vessel wall depending on the healthy or pathological state in which they take place. In this interplay, exposure to various risk factors, including high LDL-cholesterol and triglyceride plasma levels, diabetes mellitus, obesity, arterial hypertension, inflammatory mediators, and a sedentary lifestyle promotes endothelial dysfunction, thus triggering the development of cardiovascular disease.

Extracellular nucleotides, released by vascular cells and nerves surrounding the vessels, bind to cell surface receptors known as P2 receptors [11]. P2 receptors are widely distributed in the cardiovascular system whereby they regulate the functions of vascular cells through autocrine or paracrine mechanisms [11,12]. Under physiological conditions, P2 receptors signaling pathways contributes to endothelial cell homeostasis and the maintenance of some important processes such as platelet aggregation, vascular tone and proliferation of endothelial cells during angiogenesis [11,12]. Dysregulation of P2 receptor-mediated signaling could contribute to the loss of endothelial barrier integrity and affect the endothelial control of vascular tone in coronary arteries, favoring the development of cardiovascular diseases (CVD) [11,12].

Molecular cloning, pharmacological and functional data have led to P2 receptors being classified into two main families: P2X receptors (P2X-R), which are ligand-gated intrinsic ion channels comprised of homo- and hetero-oligomers that are sensitive to ATP [13], and P2Y receptors (P2Y-R), which comprise seven transmembrane domain receptors coupled to G proteins [14,15,16].

Endothelial cells express all seven P2X-R subtypes (P2X_1_-R to P2X_7_-R) [17]. In particular, P2X_4_-R has been shown to mediate endothelium-dependent vasodilatation in response to ATP released during shear stress, in addition to increasing expression of the atheroprotective gene coding for Krüppel-like factor 2 (KLF2) [18,19]. In this review, we only deal with the function of P2Y-R in the endothelium ([11,20] for reviews on P2X-R).

P2Y-R differ in their affinities for ATP, ADP, UTP, UDP, and UDP-glucose. They are expressed in diverse tissues of the body, and the affinity for a given ligand depends on the subtype, cell type, and species. Eight P2Y-R subtypes have been cloned and characterized in mammals [14,15,16]: ADP-sensitive P2Y_1_-R, P2Y_12_-R, and P2Y_13_-R; ATP- and UTP-sensitive P2Y_2_-R and P2Y_4_-R; UDP-sensitive P2Y_6_-R; ATP-sensitive P2Y_11_-R, and UDP-glucose-sensitive P2Y_14_-R. The P2Y-R subtypes P2Y_1_-R, P2Y_2_-R, P2Y_4_-R, P2Y_6_-R, and P2Y_11_-R couple mainly to G_q_, with subsequent activation of the phospholipase C pathway. P2Y_11_-R can also couple to G_s_ and stimulate adenylyl cyclase, whereas the subtypes P2Y_12_-R, P2Y_13_-R, and P2Y_14_-R couple to G_i_ to inhibit adenylyl cyclase.

Endothelial cells in culture are sensitive to gentle mechanical stimulation [21] or fluid mechanical force generated by shear stress [22,23,24,25], resulting in the release of ATP, which has a crucial role in controlling a variety of vascular functions by activating P2 purinergic receptors. A large amount of this nucleotide is released from activated/aggregating platelets during hemostasis [26] or by dying cells following medical procedures such as angioplasty or vascular stenting [23,24]. Once secreted into the extracellular medium, its concentration in plasma is locally regulated by several groups of membrane-associated ectoenzymes so that the intensity of its local effect is tightly regulated over time [27,28,29,30].

P2Y_1_-R, P2Y_2_-R, and P2Y_11_-R are abundantly expressed in mature endothelial cells isolated from umbilical cord (i.e., HUVECs) [8,17,31,32,33], which are used extensively as an in vitro model to study vascular remodeling and endothelial cell function. Additional receptors have been identified in this model, including P2Y_4_-R [17], P2Y_6_-R [17], P2Y_12_-R [31,34], P2Y_13_-R [31,35], and P2Y_14_-R [35] but little is known about their functional roles in endothelial cells. In vivo, these receptors are expressed at different levels and in different combinations depending on the vascular bed and the species studied [17,35,36,37,38].

Although the same cell type can express different P2Y-R subtypes, the activity of each subtype depends on the extracellular nucleotide and, more importantly, its concentration, which is modulated by ectonucleotidases and nucleotide translocases expressed at the cell surface [31,32,39]. P2Y-R have also been identified in several organs, and some of them are involved in blood glucose [39,40,41] and lipid metabolism regulation [42,43,44,45]. From these last locations, P2Y-R exert indirect beneficial effects on the cardiovascular system, as they contribute to the control of metabolic function such as the turnover of blood lipids and blood glucose regulation (see [11,46,47,48] for reviews).

This review discusses the role of endothelial P2Y-R in cardiovascular health, from vascular physiology to pathology.

## 2. Vascular Tone and Blood Pressure

Studies performed on cultured endothelial cells [17,23,33,34] and blood vessels isolated from rodents [23,38,49,50] identified two main receptors, P2Y_1_-R and P2Y_2_-R, that are highly expressed in a basal state in endothelium and involved in the control of endothelial nitric oxide (NO) release. These two P2 receptors play important roles in the control of vascular tone [23,38,40,50,51] (Figure 1 and Table 1). Furthermore, P2Y_2_-R has been shown to be involved in the control of blood pressure under the influence of fluid shear stress exerted by the flowing blood [23].

Elevated blood pressure is a primary risk factor for CVD, including myocardial infarction and stroke. The endothelium senses flow-induced shear stress and releases a variety of vasoactive factors to adapt vessel diameters to blood flow, which contributes to vascular tone and blood pressure regulation. Among these factors, NO plays a critical role. It is released in response to fluid shear stress exerted on endothelial cells, leading to a transient increase in the intracellular calcium concentration in endothelial cells and Ca^2+^/calmodulin-dependent activation of endothelial nitric oxide synthase (eNOS). Sustained activation of NO formation in response to shear stress requires phosphorylation of eNOS at serine 1177 [52] (or at serine 1179 [53]) by AKT kinase, although other protein kinases may also be involved (such as PKA) [54]. This mechanism enhances the enzymatic activity and alters the sensitivity of the enzyme to calcium, resulting in maximal activity at sub-physiological concentrations of calcium [52]. This results in a two- to four-fold enhancement in NO production over basal values, and this is maintained as long as the stimulus is applied [49]. NO diffuses from the endothelium and activates the soluble guanylyl cyclase located in smooth muscle cells, resulting in the production of cGMP in the media adjacent to arteries. This effect induces arterial vasorelaxation in part by decreasing the intracellular concentration of free calcium [50].

Several mechanotransducers have been shown to be involved in fluid shear stress-induced endothelial effects (mechanosensitive Piezo ion channels [2], endothelial glycocalyx layer [51], the primary cilium [55], platelet endothelial cell adhesion molecule-1 (PECAM-1) [49,56], and the G-protein-coupled receptors (GPCR), which are reviewed elsewhere [57]). Among GPCR, Wang et al. [23] have reported the involvement of endothelial P2Y_2_-R in shear stress-induced eNOS activation. Their study on bovine aorta endothelial cells (BAECs) showed that a considerable amount (approximately 8 nmol per 10^6^ cells) of the P2Y_2_-R agonist ATP is released in response to the flow-exerted laminar shear stress at 20 dynes/m^2^ compared to static flow. Moreover, siRNA knock-down studies of cultured endothelial cells (BAECs [23] or human umbilical artery endothelial cells (HUAECs) [58,59]) have shown that the mechanosensor Piezo1 mediates ATP release in response to both laminar [23,58] or oscillatory (or disturbed) [59] flow. This ATP secretion under the action of flow-induced activation of P2Y_2_-R [23,60] led to phosphorylation of AKT and eNOS, with increased NO release being detected in the culture medium [23]. All of these effects could be strongly reduced by the addition of apyrase (an ATP-degrading enzyme) to the culture medium. Transfection studies of cultured endothelial cells (BAECs) using siRNA directed against P2Y_2_-R or G_αq_/G_α11_ showed that P2Y_2_-R and G_αq_/G_α11_ are required for activation of the mechanosensory complex consisting of PECAM-1, VE-cadherin (vascular endothelial cadherin), and VEGFR-2 in endothelial cells. However, P2Y_2_-R and G_q/11_ acted upstream of the mechanosensory complex to mediate flow-induced AKT and eNOS phosphorylation in endothelial cells exposed to laminar low (HUVECs, 12 dynes/cm^2^) or high (BAECs, 20 dynes/cm^2^) shear stress, respectively [23]. By performing pressure myography studies on mesenteric arteries isolated from mice deficient in endothelium for P2Y_2_-R or G_q__/11_, and, by also performing telemetry blood pressure recording on these same mice [23], it was shown that this pathway has a critical role in the control of flow-induced vasodilatation and also blood pressure in vivo [23] (Figure 1).

P2Y_1_-R is also involved in shear stress-induced NO generation in the context of endothelial cell metabolism involving glucose transporter type 1 (GLUT1) protein expression in response to endothelial cell autophagy, which is a physiological process that is activated by shear stress [61]. Using cultured endothelial cells (BAECs), Bharath et al. [61] showed that exposure to shear stress (20 dynes/cm^2^) for three hours induced autophagy, and this metabolic process induced an increase in GLUT1 protein expression at the plasma membrane of the endothelial cells. In this experimental work, GLUT1 protein expression promoted endothelial cell glucose uptake and, therefore, ATP production through the glycolytic pathway. Using genetic (siRNA method) or pharmacological tools to study the P2Y-R involved, the authors showed that the increase in extracellular ATP in response to shear stress, via its metabolic product ADP, induced activation of P2Y_1_-R, which activated the downstream serine-threonine protein kinase δ (PKCδ) by phosphorylation of the enzyme on threonine 505. Da Silva et al. previously demonstrated this activation (PKCδ isoform) in endothelial cells (HUVECs) cultured under static conditions and exposed to ATP or UTP, but their study did not identify the purinergic receptor involved [62]. Shear stress induced autophagy and, therefore, activation of P2Y_1_-R and the downstream PKCδ enzyme, resulting in a cascade of events that leads to eNOS phosphorylation and NO production in endothelial cells (Figure 1). All of these cellular effects could be blunted in endothelial cells exposed to shear stress by transfection with an siRNA targeting Atg3, which is a key autophagy protein. This genetic inhibition of autophagy repressed both the increase in extracellular ATP and GLUT1 protein expression and thereby the phosphorylation/activation of PKCδ and eNOS, thus negating NO formation.

**Table 1 ijms-23-05883-t001:** Functions of P2Y receptors in endothelial cells and vessels.

P2Y-R Subtype (Agonist, G Protein)	Endothelial Cell Type or Tissue and Species	Function	References
**P2Y_1_** (ADP, G_q_)	BAECs	Endothelial NO release	[61]
	HUVECs	Endothelial NO release	[8]
		Endothelial cell proliferation	[31]
		Endothelial cell migration	[32]
	Mouse lung endothelial cells	Increase in ICAM-1, VCAM-1, and P-Selectin, Leucocyte recruitment and transmigration	[7]
	Mouse mesenteric venules, femoral artery	Leucocyte recruitment (in vivo) *	[7]
	Entire aorta, aortic sinus	Development of atherosclerosis lesions *	[63]
	Mouse aorta	Endothelial NO release, vasorelaxation	[8,36]
	Mouse femoral artery	Vasodilatation NO-dependent (in vivo)	
	Human veins (umbilical and chorionic)	Vasorelaxation	[8]
**P2Y_2_** (ATP = UTP, G_q/11_)	HUVECs	Endothelial NO release	[23,60]
		Endothelial cell migration	[64]
		Endothelial cell sprouting, vascular network formation	[65]
		Cytoskeletal rearrangement, mechanical properties, cell alignment under shear stress	[66,67]
	BAECs	Endothelial NO release	[23]
	HUAECs	Endothelial NOS activation under high laminar flow	[59]
		NF-κB activation, VCAM-1 expression under oscillatory flow	[59]
	Mouse aorta	Vasorelaxation	[68,69]
	Mouse mesenteric artery	Vasorelaxation NO-dependent—blood pressure (in vivo)	[23]
	Mouse entire aorta, aortic sinus	Development of atherosclerosis lesions (in vivo)	[69]
	Mouse left common carotid artery	Intimal hyperplasia (in vivo)	[59]
**P2Y_4_** (UTP = G_q_)	HUVECs	Endothelial cell migration *	[64]
**P2Y_12_** (ADP = Gi)	BAECs	apoA-I and HDL transcytosis	[34]
	HUVECs	Endothelial cell proliferation **	[70]
	HMEC-1	Reduction in the endothelial production of thrombospondin-1 **	[71]

Endothelial cell; HUVEC: human umbilical vein endothelial cell; NF-κB: nuclear factor-kappa B. * The direct role of the P2Y^1^-R in endothelium remains to be clarified. ** function based on results witth P2Y_12_-R antagonists.

Metabolic products released from organs or cells can interact with the vascular system and, according to the receptor involved, participate in the regulation of vascular tone and blood pressure via activation of a purinergic receptor downstream of the initial activated receptor. Indeed, our group [8,9,28] and others [34], using a pharmacological and genetic approach, have demonstrated that apolipoprotein A-I (apoA-I), the main apolipoprotein from high-density lipoprotein (HDL), is able to activate an endothelial P2Y-R downstream of ecto-F_1_-ATPase, which is an apoA-I receptor [34,72] anchored to the cell membrane of endothelial cells [9,34]. Our group [9,73] demonstrated for the first time that binding of apoA-I to the β subunit of ecto-F_1_-ATPase activates its hydrolytic activity and the subsequent hydrolysis of extracellular ATP into ADP by the ectoenzyme (Figure 1). The newly generated ADP binds to and stimulates a P2Y-R sensitive to ADP, the concentration of which increased in the culture medium. By using siRNA or the P2Y_1_-R antagonist MRS2179, we showed that apoA-I stimulated the downstream target P2Y_1_-R [8,31], which then induced endothelial NO synthase activation and thus also NO production. This mechanism was shown in cultured endothelial cells (HUVECs) and freshly isolated wild-type mouse aorta. Cell signaling studies using specific inhibitors and siRNA transfection of cultured endothelial cells (HUVECs) have shown that apoA-I results in activation of the downstream targets P2Y_1_-R and phosphoinositide 3-kinase β (PI3Kβ) [31], as well as phosphorylation/activation of AKT [31] on serine 473. Ex-vivo, activation of ecto-F_1_-ATPase by apoA-I has been shown to induce vasorelaxation of human veins [8], an effect that could be inhibited by the addition of NOS inhibitor L-N^G^-nitroarginine methyl ester (L-NAME). Moreover, in vivo, blood infusion of apoA-I in awake wild-type mice increased arterial femoral blood flow, and this effect could be prevented by the addition of the NOS inhibitor L-NMMA to the apoA-I infusion [8]. In addition, the increase in the arterial femoral blood flow in response to apoA-I could be inhibited by the addition of MRS2179 to the apoA-I infusion [8], suggesting a vasorelaxing effect of ADP through the activation of P2Y_1_-R located on the endothelium of blood vessels (Figure 1). To confirm this effect, the ADP analog 2-methylthioadenosine-5′-diphosphate (2-meSADP) [74] was infused in another set of wild-type mice, and an increase in blood flow occurred that could be inhibited by the addition of L-N^G^-monomethyl arginine (L-NMMA) or MRS2179 to the 2-meSADP infusion [8]. In vivo, this inhibition of increased arterial blood flow by L-NMMA, in response to the infusion of 2-meSADP, suggests that the vasodilator effect of ADP is NO-dependent.

The contribution of nucleotides to endothelium-dependent vasorelaxation and the resulting blood pressure is tightly regulated by ecto-nucleotidases located in the vessel wall. These enzymes regulate local concentrations of nucleotides and hence play a key role in regulating endothelium-related functions as well as vascular tone [36] and blood pressure [75]. Modulation of the expression or activity of these enzymes appears to occur in pathological conditions [76,77,78,79,80], and this raises the in vivo relevance of such enzymatic control in physiological and pathological conditions.

## 3. Endothelial Cell Migration and Proliferation

Nucleotides have a mitogenic action on the endothelium [81,82]. A set of studies performed on cultured endothelial cells have highlighted a role for P2Y_1_-R [31,32] and also P2Y_2_-R [64,65] in nucleotide-mediated effects on the proliferation, migration, and spreading of endothelial cells (Figure 2 and Table 1). These mechanisms represent crucial steps in wound healing and the process of vessel outgrowth such as angiogenesis, and some of them are discussed below.

The addition of ADP or 2-meSADP to cultured human umbilical vein endothelial cells (HUVECs) has been shown to stimulate endothelial cell migration under static flow [32]. This promigratory effect observed by Boyden chamber assay, as well as with wound healing assays in vitro, involves activation of the mitogen-activated protein kinase (MAPK) pathways downstream P2Y_1_-R, and it could be inhibited by the addition of MRS2179 to the culture medium. Mechanistically, MRS2179 blocked phosphorylation/activation of ERK1/2 (extracellular signal-regulated kinase), p38 kinase, and JNK (c-Jun N-terminal Kinase), resulting in inhibition of endothelial cell migration [32] (Figure 2). Moreover, the blockade of each effector separately by a specific inhibitor inhibited or decreased endothelial cell migration in the presence of ADP or 2-meSADP. In this study, the addition of AR-C69931MX (a P2Y_12/13_ antagonist) had no effect.

HDL and also apoA-I have important beneficial effects on endothelium that are related to their cardioprotective effects [83,84]. Early in vitro studies reported that both HDL and apoA-I stimulate endothelial cell proliferation [9,85,86] and migration [87,88] in addition to inhibiting apoptosis [9,89]. In vivo, these mechanisms are important for reendothelialization [87,88]. The receptors and the cell signaling pathways involved in these effects are not well understood. In this field of search, our group showed that the addition of apoA-I to cultured HUVECs stimulated their proliferation [9,31]. This effect was demonstrated to be mediated by the P2Y_1_-R [31] downstream of activation of the ecto-F_1_-ATPase by apoA-I, as it could be inhibited by the addition of MRS2179 to the culture medium. Moreover, through endothelial cell (HUVECs) transfection studies with siRNA or by using specific inhibitors, it was shown that apoA-I activates PI3Kβ and, therefore, AKT, which is activated by phosphorylation of serine 473. This cell signaling was shown to mediate endothelial cell proliferation in response to apoA-I, as the addition of the specific PI3Kβ inhibitor TGX-221 inhibited the effects of apoA-I on endothelial cell proliferation (Figure 2).

Using endothelial cells (HUVECs) cultured in a modified Boyden chamber and subjected to static flow, Kaczmarek et al. showed that the addition of ATP or UTP induced an increase in endothelial cell migration, primarily through P2Y_2_-R activation [64]. Mechanistically, P2Y_2_-R signaling pathway increased the intracellular calcium concentration from cytoplasmic stores and caused focal adhesion kinase (FAK) phosphorylation on tyrosine 397. In this set of experiments, endothelial cell migration could be attenuated by the addition of a PI3K inhibitor (wortmannin or LY294002) or an intracellular chelator of calcium ions (BAPTA-AM). BAPTA-AM also prevented phosphorylation of FAK on tyrosine 397. In addition, P2Y_2_-R activation in endothelial cells caused change in both the stiffness and adhesiveness of the plasma membrane [67] and cytoskeletal rearrangements, as shown in endothelial cells by an increase in stress fiber formation [64]. Exposure to ATP or UTP also led to an increase in α_v_ integrin expression at the protein level, which has previously been reported to play a role in mediating cell adhesion to the extracellular matrix, cell migration, and angiogenesis. ATP/UTP-sensitive P2Y_2_-R and possibly UTP-sensitive P2Y_4_-R were proposed to be involved in the observed effects, while UDP-sensitive P2Y_6_ was excluded as its expression is relatively low in HUVECs (Figure 2).

In vivo, shear stress is necessary for timely repair of injured endothelium [85,86]. Santhanoori et al. [66] studied the role of P2Y_2_-R in endothelial repair under laminar shear stress in vitro. A monolayer of endothelial cells (HUVECs) was scratched in the direction of the laminar flow with a pipette tip and then subjected to shear stress (10 dynes/cm^2^) for six hours. In this study, both exposure of the endothelial cells to a P2Y_2_-R antagonist (such as AR-C118925) and siRNA-mediated knockdown of P2Y_2_-R impaired the shear stress-induced wound closure. To explain the mechanisms involved, the authors referred to the study of Kaczmarek et al. [64], which showed involvement of P2Y_2_-R in endothelial cell migration through activation of FAK (Figure 2). Endothelial cell migration and spreading, but not proliferation, were the major mechanisms described by Albuquerque et al. [85] in a study of the effect of different levels of laminar shear stress (3, 12, and 20 dynes/cm^2^) on wound closure both in HUVECs and HUCAECs (human coronary artery endothelial cells).

By overexpressing P2Y_2_-R in HUVEC, Mühleder et al. [65] showed that P2Y_2_-R plays a role in vascular network formation. Mechanistically, it increases spontaneous endothelial sprouting and promotes the formation of a primitive vascular network. These effects are reversed by silencing P2Y_2_-R in overexpressing cells (HUVECs) or in the presence of the selective P2Y_2_-R antagonist AR-C118925XX in the culture medium. Moreover, the data show involvement of VEGFR-2 in spontaneous endothelial sprouting of HUVECs overexpressing P2Y_2_-R because these observed effects are altered by treatment with the selective VEGFR-2 inhibitor apatinib.

Several P2Y_12_-R antagonists are used in clinical practice as antithrombotic drugs [90,91] and a number of effects on endothelial function have been investigated. A recent study investigated the impact of treatment with these compounds on the angiogenic properties of HUVECs line EA.hy926 (in vitro) [70]. The culture medium was supplemented with a level of antagonist that mimics the highest measured in vivo concentration in the serum of subjects receiving a clinical or loading dose for antiplatelet therapy during angioplasty or acute coronary syndrome. The data showed that exposure of HUVECs to P2Y_12_-R antagonists (clopidrogel, prasugel, or ticagrelor) slightly reduced their in vitro proliferation. At these concentrations, the compounds did not alter endothelial cell migration, invasiveness, or tube formation in vitro. Moreover, the endothelial cells were still able to repopulate the denuded areas in an in vitro wound healing assay in the presence of VEGF [70]. Rather, several studies reported the ability of antiplatelet P2Y_12_-R antagonists, clopidrogrel and ticagrelor, to improve endothelial functions in different clinical settings such as in patients with stable coronary artery disease (CAD) and type 2 diabetes (TD2) [92,93,94,95,96]. Interestingly, a recent study reported that the P2Y_12_-R antagonist, cangrelor, increased the production of the pro-thrombotic protein thrombospondin-1 (TSP-1) by human microvascular endothelial cells (HMEC-1) in response to adenosine (P1) receptor (A_2_AR) agonists treatment [71]. Therefore, a better understanding of the specific mechanism underlying TSP-1 production by endothelial cells in the context of P2Y_12_-R inhibition and A_2_AR activation is important for antiplatelet therapies.

Endothelial cell stimulation in vitro with UTP or ATP or the corresponding nucleoside diphosphate forms (ADP or UDP) is associated with endothelial secretion of proangiogenic factors [97] and also modulation of cell surface receptor expression, with—for example—an increase in some P2Y-R subtypes [97]. The physiological relevance of these biological effects in vivo still needs to be elucidated in the setting of vascular pathology, where angiogenesis or vascular repairs come into play.

In conclusion, certain purinergic P2Y-R have angiogenic properties that have been well studied in vitro. These observations remained to be reproduced in vivo, under physiological or pathological conditions, to better understand how they are involved in the studied effects and according to the vessels and the environment in which the mechanisms are studied (inflammation, oxidizing stress, high-pressure). These effects play a crucial role in re-endothelialization after coronary angioplasty. They also play a role in maintaining the endothelial barrier, which prevents any protein or fluid extravasation, and thrombosis under normal and pathological conditions (vascular injury, atherosclerotic plaque rupture). This mechanism also contributes to the growth of new vessels in an ischemic limb or organ, and also after arterial occlusion [98,99].

## 4. Vascular Inflammation and Atherogenesis

Nucleotides are implicated in the inflammatory response [100], and they exert a direct action on circulating blood cells and vascular cells such as endothelial and smooth muscle cells. On the vessel wall, they have long-term trophic effects on cell growth and proliferation or death [82]. The role of P2 receptors in the development of vascular inflammation and atherosclerosis has not been investigated much to date, and recent advances are reviewed elsewhere [11,39,52,101]. One of the key events in the physiopathology of atherosclerosis is exposure of adhesion molecules on endothelium in response to inflammatory cytokines to recruit subsets of leukocytes [102]. This mechanism promotes monocyte adhesion to endothelium and their transmigration into the subendothelial space. Recent studies involving genetically modified mice have highlighted a role for P2Y_1_-R [7,63] and also P2Y_2_-R [28,74,103] in endothelium in regulation of the expression of these adhesion molecules and also the progression of atherosclerosis. Some of these studies are discussed below.

In a mouse model of acute vascular inflammation and deficient for P2Y_1_-R in the whole body (P2Y_1_-KO), a reduction in leucocyte recruitment was shown in the femoral artery after local subcutaneous injection of both tumor necrosis factor-α (TNF-α) and interleukin 1β (IL-1β), two inflammatory cytokines [7]. This observation was reproduced in mice lacking apolipoprotein-E (ApoE-KO) by chronic infusion of the P2Y_1_-R antagonist MRS2500 for the three days with osmotic pumps implanted in the subcutaneous dorsal space. P2Y_1_-R is present on blood cells (platelets, monocytes/macrophages, and lymphocytes) [11,104] and other cells of the vessel wall (endothelial cells and fibroblasts) [11,105]. On platelets, this receptor is necessary for normal platelet activation and aggregation in response to ADP [26]. However, in the circulatory system, activated platelets interact with leucocytes and promote monocyte adhesion to endothelium, a mechanism involved in atherosclerosis development [102,106]. To decipher the role of P2Y_1_-R expressed on blood cells from other cells in the body in vascular inflammation, bone marrow transplantation studies using apoE-KO mice and P2Y_1_/apoE double knock-out mice have been performed. In these two types of transplanted mice, leucocyte recruitment in the femoral artery was studied after subcutaneous injection of the inflammatory cytokines TNF-α and IL-1β. These experiments allowed exclusion of a role of P2Y_1_-R in blood cells in vascular inflammation. Subsequently, using endothelial cells isolated from murine lung to study their interaction with monocytes isolated from the blood of wild-type or P2Y_1_-KO mice, it was shown that P2Y_1_-R in endothelium plays a role in leucocyte adhesion and transendothelial migration induced by TNF-α [7]. This was ascertained by flow cytometry experiments on cultured endothelial cells, showing an increase in intercellular adhesion molecule-1 (ICAM-1), VCAM-1, and P-selectin on endothelium following TNF-α exposure for the wild-type group, while there was a reduction at the surface of the P2Y_1_-R-deficient endothelial cells. The mechanisms and cell signaling involved in TNF-α-induced P2Y_1_-R activation and the expression of adhesion molecules downstream of P2Y_1_-R have, however, not yet been elucidated.

By studying early and late stages of the development of atherosclerotic lesions in an atheroprone mouse model (apoE-KO) and deficient for the P2Y_1_-R in the whole body, Hechler et al. [93] showed a reduction in the size of atherosclerotic lesions at 17 weeks in the entire aorta and at 30 weeks in the aortic sinus in these mice maintained on a standard low-fat chow diet. Immunohistochemical analysis of atherosclerotic plaques from aortic sinus isolated at 30 weeks showed a reduction in VCAM-1 staining, less macrophage infiltration, and reduced smooth muscle cell accumulation [63] compared to the apoE-KO group. In this experimental mouse model, deficient for both apoE and P2Y_1_-R in the whole body (double knock-out), bone marrow transplantation experiments ruled out a role for P2Y_1_-R expressed in blood cells in atherosclerotic lesion development. The Gachet group discussed a potential role for P2Y_1_-R in liver and/or the vasculature in the observed phenotype [63]. Although the contribution of endothelial P2Y_1_-R is well established in endothelium-dependent vasodilatation, its role in vascular inflammation and atherosclerosis development remain to be fully demonstrated.

Using a mouse model specifically lacking P2Y_2_-R in the endothelium, the Seye group demonstrated a contribution of this endothelial cell receptor to the pathogenesis of atherosclerosis [69]. In these mice, myography studies performed on aortic rings revealed impairment of the endothelium-dependent relaxation induced by the addition of ATP or UTP. Moreover, endothelial P2Y_2_-R deficiency was associated with attenuation of eNOS expression and activity in mouse aorta, but no change in blood pressure or the heart rate could be discerned by radio-telemetry in this mouse model. The role of endothelial P2Y_2_-R in atherosclerosis was further studied in endothelial-specific P2Y_2_-R-deficient mice on an apoE-KO genetic background that were maintained on a standard chow diet for 25 weeks [69]. Immunohistology studies performed on the aortic sinus and entire aorta showed a reduction in atherosclerotic lesions compared to the control mice, which did not lack endothelial P2Y_2_-R. In addition, analysis of the cellular composition on cross-sections of aortic sinuses revealed a reduction in VCAM-1 staining in endothelium lacking P2Y_2_-R, and this was associated with a decrease in monocyte infiltration in the lesion area, as evidenced by a decrease in Mac-3 staining. Moreover, endothelial cell P2Y_2_-R deficiency induced an increase in smooth muscle cells and collagen content in atherosclerotic plaques, resulting in the formation of a subendothelial fibrous cap with increased plaque stability.

In conclusion, a few studies to date, limited to mouse models, have investigated the selective contribution of endothelial P2Y-R subtypes to vascular inflammation and atherosclerosis.

## 5. HDL Transcytosis in Endothelial Cells

Ten years ago, Von Eckardstein’s group showed a role for endothelial P2Y-R in both HDL and apoA-I (the major HDL protein) transport through aortic endothelial cells [84,107] (Table 1). This mechanism appears to be atheroprotective, as it provides access to foam cells located within the arterial wall. Indeed, at this location, apoA-I or HDL mediate cholesterol efflux from cells, which is the first step of a process called reverse cholesterol transport (RCT) [101].

Several receptors present in endothelium have previously been reported to be involved in this transport. Using siRNA silencing, a study on bovine aortic endothelial cells (BAECs) has shown that the ATP-binding cassette transporter A1 (ABCA1) binds to apoA-I and contributes to lipid-free apoA-I endocytosis and transcytosis across the endothelial barrier [108], while other receptors such as ABCG1 and the scavenger receptor BI (SR-BI) have been shown to be involved in binding, endocytosis, and transcytosis of mature HDL [103]. Subsequently, the role of ecto-F_1_-ATPase, an apoA-I receptor, in mediating the transendothelial transport was studied, again using BAECs [34]. The addition of apoA-I to the culture medium specifically stimulated ecto-F_1_-ATPase activity. The generated extracellular ADP, as well as apoA-I, stimulated the binding, cell association, and internalization of HDL in endothelial cells. These steps could be mimicked by the addition of ADP to the culture medium and inhibited in the presence of the P2Y_12_-R inhibitor 2-methylthioadenosine 5′-monophosphate (2-meSAMP). The addition of the specific P2Y_1_-R inhibitor MRS2179 did not have any effect. The association and transport of apoA-I or HDL could also be inhibited by ATPase inhibitory factor 1 (IF1), which is a specific ecto-F_1_-ATPase inhibitor [109].

Ecto-F_1_-ATPase/P2Y-induced HDL endocytosis is a process that was previously identified by our group in hepatocytes [29,44,73,110,111,112,113,114]. Our study focused on characterization of both cellular events and cellular signaling downstream of the purinergic P2Y_13_-R in hepatocytes. Our group also investigated to what extent this pathway participates in RCT and protection offered against atherogenesis [43,44,45,110].

These data showed a role for endothelial P2Y_12_-R in the regulation of endothelial cell permeability to macromolecules such as apoA-I and HDL. However, endothelial cells in different vascular beds are heterogenous and, therefore, this process may concern certain vascular beds where, for example, the endothelial cell barrier is continuous. Moreover, this transport may be locally modulated by factors released by neighboring cells or other processes such as shear stress [115]. The cell signaling pathway involved in the events downstream of the activation of P2Y_12_-R has not been identified to date. The extent to which this process is functional in vivo, in particular in the setting of endothelial dysfunction or atherosclerosis, remains to be unraveled.

## 6. Fluid Shear Stress-Induced Change in Endothelial Phenotype, Vascular Remodeling, and Atherogenesis

In the circulatory system, endothelial cells form a dynamically mutable interface in contact with biochemical stimuli and hemodynamic forces. These cells sense mechanical forces such as shear stress and adapt their phenotype to blood flow [116]. Indeed, in vascular regions where shear stress is laminar, endothelial cells elongate and align in the direction of flow [85]. This is accompanied by a change in cytoskeletal organization, with the assembly of stress fibers in the direction of flow, a process that increases the resistance of cells against shear stress [85]. This endothelial phenotype contrasts with that observed in regions where blood flow is disturbed, such as arterial bends and bifurcations, which are naturally atheroprone regions. At these locations, endothelial cells display a polygonal or cobblestone morphology [117]. The molecular mechanisms involved in endothelial cell phenotype change according to the coordinated action of blood flow and shear stress are not well understood. Recent advances from studies of the interaction between purinergic receptors sensitive to ATP released upon blood flow stimulation and cytoskeletal rearrangement have highlighted a potential role for some P2Y-R.

Indeed, upon blood flow stimulation, endothelial cells release nucleotides that, with the coordinated actions of endothelial P2Y-R and other mechanoreceptors, activate cell signaling pathways involved in cytoskeletal rearrangement, vasodilatation, and vascular remodeling. A study of the role of P2Y_2_-R in cytoskeletal alterations in response to shear stress in cultured endothelial cells (HUVECs) found that six hours of laminar fluid shear stress (10 dynes/cm^2^) induced an increase in P2Y_2_ mRNA expression [66], an effect that was shown to influence both cell elongation and alignment [66] compared to cells placed under a static flow and that exhibited a cobblestone morphology with no defined orientation. These effects were impaired by the addition of a P2Y_2_-R antagonist to the culture medium (AR-C118925), and also by transfection of the cells with an siRNA directed against P2Y_2_-R. In this cell model, cell signaling studies performed on cell lysates involving immunoblotting with various antibodies have shown that shear stress increases the phosphorylation of FAK at tyrosine 397 and its target cofilin-1, two proteins involved in cytoskeletal reorganization. The data also showed that there was an increase in the phosphorylation of AKT and eNOS, which are involved in vasodilatation. These changes were dependent on P2Y_2_-R, as they could be reduced by P2Y_2_-R knockdown. Moreover, they appear to be driven by the Arg-Gly-Asp (RGD) sequence of the integrin-binding domain of P2Y_2_-R, as the observed change in cell morphology and protein activation under shear stress were abolished or decreased in cells expressing a mutation in the RDG sequence of the integrin-binding domain of P2Y_2_-R.

The endothelium also plays a crucial role in the control of vascular remodeling. In direct contact with the blood, it senses the mechanical forces elicited by an increase or decrease in blood flow, thereby triggering biochemical events to adapt the vascular tone and vessel structure to local variations in flow and pressure. The mechanisms by which the endothelium controls vascular remodeling are not yet known. Neointimal hyperplasia develops at locations where shear stress is low or after arterial injury of the endothelium. This vascular remodeling contributes to atherosclerosis development and restenosis after angioplasty [118]. Migration and proliferation of vascular smooth muscle cells are the underlying causes, and these events are preceded by an influx of leucocytes into the intima. Nitric oxide (NO), which is a potent vasodilator produced by endothelium in response to an increase in flow (or shear stress), and which mediates anti-atherogenic effects, has been shown to play a protective role in vascular remodeling. It diffuses into the vessel wall and inhibits the proliferation and migration of cultured vascular smooth cells [119,120]. In vivo, it has been shown to inhibit neointima formation after vessel injury in rodents [121,122,123].

In order to understand the mechanisms involved in neointima development following an arterial injury, different approaches to generate vascular stress have been applied in rodents, with interpretation of the mechanisms depending on the model used, the experimental approach, and the presence of other predisposing factors. The experimental techniques include balloon injury, endothelial denudation, partial or complete flow obstruction, and advential cuff placement. Some of these approaches are presented and discussed elsewhere [124]. In order to discuss recent advances concerning the role of P2Y-R in endothelium and vascular remodeling, we have focused on an experimental model whereby a disturbance near the carotid bifurcation results in induction of a local hemodynamic change that facilitates vascular remodeling in the left common carotid artery. The perturbation consists of performing ligation of the ipsilateral external carotid artery near the carotid bifurcation, which reduces the blood flow (or-shear stress) in the left common carotid artery. A decrease in vascular diameter of the common carotid artery is expected to develop two weeks after the surgery and accompanies the chronic flow reduction [124,125]. Mechanistically, this effect is dependent upon the presence of vascular endothelium [125,126]. In addition, this ligation—and this can also be observed following a total ligation of the left common carotid artery near the bifurcation [127]—induced thickening of the vascular wall with the development of intimal hyperplasia [124,127]. Nam et al. were the first to describe a partial carotid ligation model to link these changes in the intima to atherogenesis [128]. The surgical technique induced a low and disturbed flow in the left common carotid artery that caused endothelial dysfunction and rapid atherosclerosis development after two weeks in apoE-KO mice fed a high-fat diet.

Several studies have provided evidence of a role for extracellular nucleotides in the development of intimal hyperplasia. This is supported by studies showing higher mRNA expression of P2Y_2_-R on the luminal edge of rat aortic neointima [129] at 8 and 20 days after endothelial denudation by use of a balloon catheter. Moreover, it was observed that the local delivery of nucleotides via a mini-osmotic pump to the collared rabbit carotid artery accelerated lesion development [130]. By contrast, intimal thickening was reduced in mice deficient in CD39 (Cd39^_^KO) that underwent endothelial denudation on the left common carotid artery with an angioplasty guidewire [131]. CD39 is a dominant vascular nucleoside triphosphate diphosphohydrolase (NTPDase) localized at the membrane of endothelial cells [132] and vascular smooth muscle cells [131]. The ectoenzyme, also called apyrase, hydrolyses nucleoside tri- and diphosphates, thereby influencing the levels of extracellular nucleotides [27,28]. Among the various possible mechanisms, the authors suggested desensitization of some P2Y-R in response to an excess of extracellular nucleotides released following the injury. This effect locally modulates cellular functions and, therefore, inhibits neointima formation. The contribution of P2Y_2_-R to intimal hyperplasia had been previously studied in a mouse model where the receptor was eliminated in the body by genetic deletion (P2Y_2_-KO) and bearing a cuff placed on the femoral artery for 14 days [133]. Under these experimental conditions, P2Y_2_-R deletion induced a decrease in intimal hyperplasia, whereas a dramatic increase in intimal lesions was shown in transgenic rats overexpressing P2Y_2_-R. In vivo, experiments examining smooth muscle cell proliferation and migration from the media to the intima, and also monocyte migration and invasion in the intima, showed a crucial contribution of P2Y_2_-R in these mechanisms [130,133]. Previous studies have shown involvement of P2Y_2_-R expressed in smooth muscle cells in regulation of smooth muscle cell proliferation and migration [134]. However, the specific role of P2Y_2_-R in endothelium in these processes has not yet been investigated. Recently, a group studied the role of the endothelial P2Y_2_-R that couples to the G proteins Gα_q_/Gα_11_ in vascular remodeling [23]. They performed experiments on mice with endothelium deficiency for the subunits Gα_q_/Gα_11_. In these mice, ligation of the left external carotid artery did not induce a decrease in the diameter of the left common carotid artery, as seen previously with wild-type mice [126]. This observation is in accordance with a previous study that demonstrated a role for eNOS in vascular remodeling that was lost in mice lacking eNOS [126]. However, a strong reduction in neointima formation was observed after ligation of the left common carotid artery near the bifurcation in endothelium-specific Gα_q_/Gα_11_-deficient mice. From these experiments, it would appear that endothelial Gα_q_/Gα_11_ –mediated signaling plays a role in vascular remodeling. More convincingly, the group studied the role of endothelial P2Y_2_-R and these subunits in atherogenesis. Using a mouse model prone to atherosclerosis [59] (LDLr-KO mouse) and fed a high fat diet, and by performing a partial ligation of both the left internal and external carotid arteries as previously described [101], a strong decrease in intimal lesions was shown to occur in the mice with endothelial-specific deficiency for Piezo1, P2Y_2_, or Gα_q_/Gα_11_ compared to control LDLr-KO mice. These last data show for the first time a contribution of endothelial P2Y_2_-R in neointima formation and atherogenesis.

There is increasing evidence that high laminar flow and disturbed flow induce different signal transduction processes in endothelial cells that result in anti- and pro-atherogenic phenotypes, respectively [65,135,136]. Disturbed flow has been shown to promote inflammatory signaling pathways in cultured endothelial cells, with activation of nuclear factor-kappa B (NF-κB), which in endothelium increases the expression of leucocyte adhesion molecules (Vascular cell adhesion protein 1, VCAM-1) and chemokines (monocyte chemoattractant protein 1, MCP1) [59,137]. By contrast, high laminar shear stress has been shown to induce atheroprotective signals in cultured endothelial cells, including eNOS activation [59]. Understanding the mechanisms of endothelial phenotypic modulation that lead to a dysfunctional state, and the predisposing factors, is the goal of concerted research efforts regarding vascular diseases and their treatment. In this research area, Albarran-Juarez et al. studied the role of the pathway consisting of the mechanosensitive cation channel Piezo1, the purinergic receptor P2Y_2_, and G_q_/G_11_-mediated signaling in flow pattern-induced atheroprotective signals (eNOS activation) or inflammatory signals (VCAM-1 expression, NF-κB activation) in vitro in HUAECs (human umbilical artery endothelial cells) after transfection with siRNAs directed against P2Y_2_-R, Piezo 1 or Gα_q/_G_11_ [59] The role of this pathway in atherosclerosis development has consequently been investigated in LDL-r KO mice fed a high-fat diet for 4 to 16 weeks, and these mice were deficient for endothelial Piezo1, P2Y_2_-R, or Gα_q_/Gα_11_. In vitro, oscillatory flow (4 dynes/cm^2^) increased ATP secretion by endothelial cells, which induced NF-κB activation and its nuclear translocation as well as VCAM-1 expression. NF-κB translocation and VCAM-1 expression could be inhibited by siRNA knock-down of Piezo1 or Gα_q_/Gα_11_ subunits. HUAEC exposure to high laminar flow (15 dynes/cm^2^) for 15 min was shown to increase eNOS phosphorylation at serine 1177, whereas no activation of NF-κB was seen. By contrast, exposure to high-frequency oscillatory flow (15 dynes/cm^2^) for 15 min did not affect eNOS phosphorylation, but it did increase NF-κB activation, and this was associated with an increase in IκBα degradation based on immunoblotting results. These effects have been replicated and studied in endothelial cells (HUAECs) transfected with siRNAs directed against Piezo1 or the Gα_q_/Gα_11_ subunits, and in these transfection studies, the siRNA-mediated knock-down inhibited the observed effects on eNOS phosphorylation or NF-κB activation, depending on the flow pattern. The physiological relevance of these observations is that, in vivo, there was a decrease in atherosclerotic plaque on the entire aorta, the aortic valve region, and also the brachiocephalic arteries in LDL-r KO mice deficient for Piezo1 or G_αq_/G_α11_ subunits and fed a high-fat diet for 16 weeks compared to the LDLr-KO control group.

To summarize, this series of experimental investigations has shown a differential role for the endothelial P2Y_2_-R in response to hemodynamic forces that can be encountered at certain locations of the arterial vessel. This physiological adaptation can initiate cardiovascular disease, especially if the cells are exposed to pro-inflammatory or atherogenic factors. Understanding this phenotypic modulation will assist with the design of better pharmacological drugs in the field of metabolic diseases.

## 7. Discussion: Pharmacological Approaches Targeting Endothelial P2Y-R

Preclinical studies that uncovered the role of P2Y-R in endothelium homeostasis raised interest in the development of drugs targeting a specific P2Y-R subtype for the treatment of CVD such as hypertension, thrombosis, intimal hyperplasia and atherosclerosis. In particular, several in vitro and in vivo studies support the development of compounds targeting P2Y_1_-R, P2Y_2_-R and P2Y_12_-R activities for the prevention and treatment of cardiovascular diseases (Table 1).

Several P2Y_12_-R antagonists are used in clinical practice as antithrombotic drugs [90,91] and a number of effects on endothelial function have been investigated. A recent study investigated the impact of treatment with these compounds on the angiogenic properties of HUVECs (in vitro) [70]. The culture medium was supplemented with a level of antagonist that mimics the highest measured in vivo concentration in the serum of subjects receiving a clinical or loading dose for antiplatelet therapy during angioplasty or acute coronary syndrome. The data showed that exposure of HUVECs to P2Y_12_-R antagonists (clopidrogel, prasugel, or ticagrelor) slightly reduced their in vitro proliferation. At these concentrations, the compounds did not alter endothelial cell migration, invasiveness, or tube formation in vitro. Moreover, the endothelial cells were still able to repopulate the denuded areas in an in vitro wound healing assay in the presence of VEGF [70]. Rather, several studies reported the ability of antiplatelet P2Y_12_-R antagonists, clopidrogrel and ticagrelor, to improve endothelial functions in vivo [92,93,94,95,96]. Interestingly, a recent study reported that the P2Y_12_-R antagonist, cangrelor, increased the production of the pro-thrombotic protein thrombospondin-1 (TSP-1) by human microvascular endothelial cells (HMEC-1) in response to adenosine (P1) receptor (A_2_AR) agonists treatment [71]. Therefore, a better understanding of the specific mechanism underlying TSP-1 production by endothelial cells in the context of P2Y_12_-R inhibition and A_2_AR activation is important for antiplatelet therapies.

ApoE and LDLr knock-out mice lacking P2Y_2_-R in endothelial cells are protected from atherogenesis [59,69], consistent with a role of P2Y_2_-R in promoting inflammation via NF-κB in atheroprone vascular areas exposed to disturbed flow [59]. However, endothelial P2Y_2_-R activity in mice was also reported to control flow-induced NO production, vascular tone and blood pressure [23]. Therefore, inhibition of P2Y_2_-R is not likely a good therapeutic strategy to reduce atherogenesis since it may lead to increased vascular tone and blood pressure. To date, two P2Y_2_-R agonists, denufosol and diquafosol, have been approved for the treatment of cystic fibrosis and dry eyes disease, respectively [138,139]. The potential effect of these drugs in long-term cardiovascular health should be carefully scrutinized.

A dual P2Y_1_-R and P2Y_12_-R antagonist, GLS-409, was shown to attenuate platelet-mediated thrombosis in rats [140]. Given that endothelial P2Y_1_-R activity contributes NO-dependent vasorelaxation in mice and endothelial cell proliferation and migration [8,33,34] (Figure 1 and Figure 2), P2Y_12_-R antagonists treatment might favor endothelial dysfunction and hypertension.

Although other P2Y_1_-R and P2Y_2_-R agonists and antagonists have been designed [141,142,143], they were only assessed in preclinical models but not in clinical setting. Limitations of such compounds for clinical use could be poor pharmacokinetic properties and lack of target and tissue selectivity. Pharmacological properties common to RCPG should be also considered when designing compounds targeting P2Y-R, including receptor desensitization and degradation, homo- and hetero-dimerization, stimulus trafficking and biased agonism efficacy [111,135,136,144]. A better characterization of these structural and pharmacological aspects governing P2Y-R activity and function will help to discover efficient and specific compounds targeting endothelial P2Y-R for the treatment of cardiovascular disorders.

## 8. Conclusions

Endothelial P2Y-R are involved in various endothelial functions (Table 1). In association with other membrane proteins, these receptors coordinate the activation of downstream effectors that determine endothelial effects. New advances based on genetic mouse models specifically modified in terms of the endothelium have provided important new knowledge regarding the role of P2Y-R in vascular physiology and vascular diseases. As the role of endothelial P2Y-R in endothelium may change under a particular pathological state, this modification warrants being fully understood and interpreted in order to better define new pharmacological targets in vascular pathology.

## Figures and Tables

**Figure 1 ijms-23-05883-f001:**
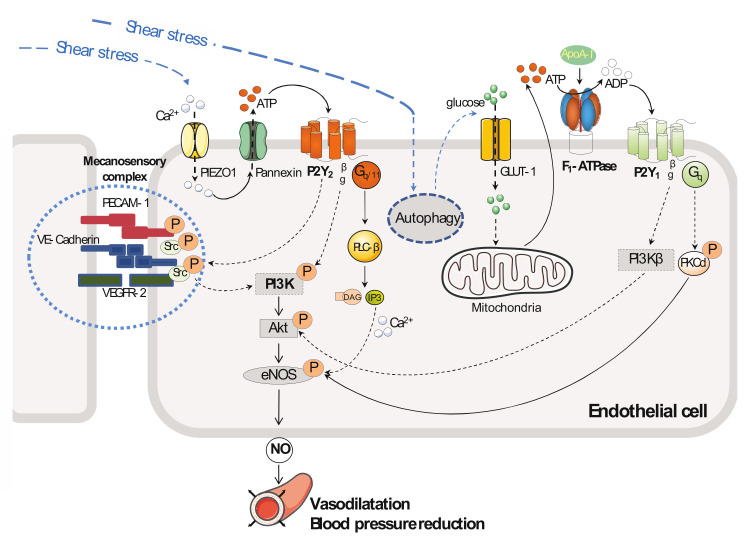
P2Y-R and vascular tone. P2Y_2_-R (**left**). Endothelial Piezo1 mediates flow-induced ATP release through pannexin channels. Extracellular ATP then activates a signaling pathway that involves the P2Y_2_-R/PLCβ/PI3K/AKT signaling cascade, leading to phosphorylation/activation of endothelial nitric oxide synthase (eNOS). P2Y_1_-R (**right**). Shear stress increases autophagy, which sequentially promotes GLUT-1 expression, cellular glucose uptake, ATP production via the glycolytic pathway, and ATP release into the extracellular medium. Extracellular ATP can be hydrolyzed to ADP by ecto-F_1_-ATPase anchored to the plasma membrane of endothelial cells. This hydrolytic activity of ecto-F_1_-ATPase is stimulated by apoA-I, which is the main HDL apolipoprotein, and the newly generated extracellular ADP sequentially activates P2Y_1_-R, PKCδ or PI3Kβ/Akt, and eNOS. Both P2Y_1_-R- and P2Y_2_-R–mediated eNOS activation lead to the production of nitric oxide (NO), which diffuses extracellularly to promote blood vessel dilatation and a decrease in blood pressure.

**Figure 2 ijms-23-05883-f002:**
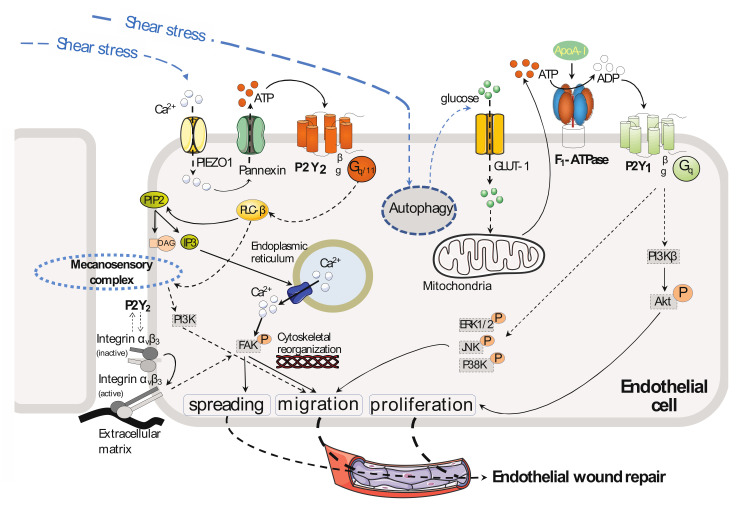
P2Y-R and endothelial cell proliferation, migration, and spreading. Endothelial cells release ATP in response to fluid shear stress, which activates purinergic receptor-mediated endothelial cell alignment, cytoskeletal reorganization, and wound repair. In particular, extracellular ATP stimulates P2Y_2_-R, which subsequently activates phospholipase C-β (PLC β), generates inositol 1,4,5-trisphosphate (IP3), increases the cytosolic Ca^2+^ concentration, and activates focal adhesion kinase (FAK), which mediate cytoskeletal reorganization and the consequent cell migration and spreading. Furthermore, extracellular ATP increases cell migration in a phosphatidylinositol 3-kinase (PI3-K)–dependent manner. ATP also induces increased expression of α_v_ integrin, which plays a role in mediating endothelial cell migration and cell alignment. Extracellular ATP is also hydrolyzed to ADP by ecto-F_1_-ATPase anchored to the plasmatic membrane of endothelial cells, a process that is stimulated by apoA-I, which is the main HDL apolipoprotein. The newly generated extracellular ADP stimulates P2Y_1_-R, which subsequently activates mitogen-activated protein kinase pathways, as evidenced by increased phosphorylation of extracellular signal-regulated kinase (ERK)1/2, c-Jun N-terminal kinase (JNK), and p38 kinase. These signaling events promote endothelial cell migration. P2Y_1_-R activation by ADP also induces activation of the PI3Kβ isoform and subsequent endothelial cell proliferation. Overall, these mechanisms contribute, in vitro, to endothelial wound repair, and they may contribute, in vivo, to reendothelialization and angiogenesis. Mechanosensory complex: PECAM-1, VE-Cadherin, VEGFR-2.

## Data Availability

Not applicable.

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
