# Peer review of "The Interplay of Endothelial P2Y Receptors in Cardiovascular Health: From Vascular Physiology to Pathology"

_ijms, 2022, doi:10.3390/ijms23115883_

Round 1
Reviewer 1 Report
The authors of the manuscript ‘Purinergic P2Y receptors in endothelium: from vascular physiology to pathology, a review based on recent advances’ describe the current understanding of the vital role of purinergic receptors in endothelial cells homeostasis. The subject is interesting and falls well within the scope of the Journal.
In principle, the manuscript is potentially interesting but I have several questions:
- The statement in the title ‘recent advances’ does not reflect the cited literature. In fact, only 22 references among 123 are from recent 5 years
- Authors do not cite very similar review published in 20220 in IJMS ‘P2Y Purinergic Receptors, Endothelial Dysfunction, and Cardiovascular Diseases’ 10.3390/ijms21186855
- Many recent references in the field are not discussed (e.g., DOI: 10.3390/ijms22020624 or 10.1016/j.mvr.2021.104218)
- I would like more information on how understanding of activation/inhibion of particular P2YRs could be useful for treating endothelial cells-related diseases. Are there any clinical trials in this field? If not, why?
Author Response
Point 1: The statement in the title ‘recent advances’ does not reflect the cited literature. In fact, only 22 references among 123 are from recent 5 years
Response 1: The title has been changed accordingly. Meanwhile, we also took in consideration suggestions from reviewer #2. The title is now: “The interplay of endothelial P2Y receptors in cardiovascular health- from vascular physiology to pathology”.
Point 2: Authors do not cite very similar review published in 2020 in IJMS ‘P2Y Purinergic Receptors, Endothelial Dysfunction, and Cardiovascular Diseases’ 10.3390/ijms21186855
Response 2: We have now cited this recent and interesting review in the Introduction section (Ref #11, Page 2, lane 48, 51 and 54).
Point 3: Many recent references in the field are not discussed (e.g., DOI: 10.3390/ijms22020624 or 10.1016/j.mvr.2021.104218)
Response 3: We have now cited those recent references (Ref #73, page 8 lane 299 and #76, page 9, lane 344)
Point 4: I would like more information on how understanding of activation/inhibition of particular P2YRs could be useful for treating endothelial cells-related diseases. Are there any clinical trials in this field? If not, why?
Response 4: We have now adressed this topic in a new Discussion section entitled “pharmacological approaches targeting endothelial P2Y-R” (page 14 lane 631).

Reviewer 2 Report
Title: Purinergic P2Y receptors in endothelium: from vascular physiology to pathology, a review based on recent advances
The authors extensively discuss the role of extracellular
nucleotides and endothelial purinergic P2Y receptors (P2Y-R) in vascular protection. The authors also discuss how the endothelium tissue plays a key role in releasing several mediators that regulate local processes and that protect against the development of cardiovascular disease. The process and pathways involved at a molecular level involved in the regeneration of the endothelial tissue are also discussed. The authors also cite various studies showing how endothelial P2Y2-R, and possibly P2Y1-R, have also been 15
reported to promote vascular inflammation and atheroma development in mouse models, with endothelial P2Y2-R also being described as promoting vascular remodeling and neointimal hyperplasia. Another interesting finding of how P2Y12-R has been found to trigger HDL transcytosis through endothelial cells is discussed in detail with interesting examples from the literature. This process is known to be protective against lipid deposition in
the vascular wall. P2Y-R receptor, its characterization and understanding its downstream signaling can be extremely beneficial for the development of a new targets for drug development.
The review article is highly comprehensive and there are a wast number of studies cited. The authors have done an excellent job at combining all the relevant information and presenting it in this manuscript. The reviewer would suggest restructuring the manuscript so that it will be viewed ad read by researchers of various backgrounds. The overall manuscript should flow from the importance of endothelial receptors in cardiovascular health in the introduction section and discussing the disease burden, then discussing all about P2Y-R receptors that the authors have already mentioned, and finally the manuscript should conclude with a discussion section discussing how P2Y-R receptors can be a novel target fort he development of various new drugs.
Although the manuscript is perfectly written, the reviewer would like to add the following suggestions to better improve the quality of the presented information
Major comments
- The discussion section needs to be incorporated. The implications of the P2Y-R receptors should be explained in more detail. The future implications should be stated, for example, what kind of drugs can be developed, and what effects will those drugs have. Here a note about the future directions about the research opportunities for P2Y-R receptors can be included. Additionally, a vision for the future, about the clinical translatable value of the P2Y-R receptors should be predicted based on the studies presented in the manuscript.
- The introduction can include a section about how endothelial receptors play an important role in cardiovascular health and how the dysregulation of some can be detrimental.
Minor comments:
- The authors should include figures (pre-clinical data figures) followed by their explanations, of the most relevant section of the P2Y-R receptors that they are discussing.
- The authors can also think of changing the title of the article to something indicating how P2Y receptors can be important for cardiovascular health. Alternatively, the title can be - The interplay of P2Y receptors in cardiovascular health- from vascular physiology to pathology, a review based on recent advances.
Author Response
Point 1: The discussion section needs to be incorporated. The implications of the P2Y-R receptors should be explained in more detail. The future implications should be stated, for example, what kind of drugs can be developed, and what effects will those drugs have. Here a note about the future directions about the research opportunities for P2Y-R receptors can be included. Additionally, a vision for the future, about the clinical translatable value of the P2Y-R receptors should be predicted based on the studies presented in the manuscript.
Response 1: We have now adressed this topic in a new Discussion section entitled “pharmacological approaches targeting endothelial P2Y-R” (page 14 lane 631).
Point 2: The introduction can include a section about how endothelial receptors play an important role in cardiovascular health and how the dysregulation of some can be detrimental.
Response 2: We have now addressed this aspect in a new paragraph in the Introduction section.
Point 3: The authors should include figures (pre-clinical data figures) followed by their explanations, of the most relevant section of the P2Y-R receptors that they are discussing.
Response 3: As the reviewer suggested, figure 1 and 2 are now presented at the beginning the related sections.
Point 4: The authors can also think of changing the title of the article to something indicating how P2Y receptors can be important for cardiovascular health. Alternatively, the title can be - The interplay of P2Y receptors in cardiovascular health- from vascular physiology to pathology, a review based on recent advances.
Response 4: Title has been changed accordingly. Meanwhile, we also took in consideration suggestions from reviewer #1. The title is now: “The interplay of endothelial P2Y receptors in cardiovascular health- from vascular physiology to pathology”

Round 2
Reviewer 1 Report
All suggestions were introduced by authors.
Reviewer 2 Report
The authors have perfectly incorporated the suggested changes. This definitely improves the overall quality of the manuscript. The reviewer would like to appreciate the authors for accepting and adding these changes to their manscript.
The manuscript can be accepted in its current form.